# Human Immunodeficiency Virus Infection-Associated Cardiomyopathy and Heart Failure

**DOI:** 10.3390/jpm12111760

**Published:** 2022-10-24

**Authors:** Aikaterini Papamanoli, Brandon Muncan, Jeanwoo Yoo, George Psevdos, Andreas P. Kalogeropoulos

**Affiliations:** 1Internal Medicine, Zucker School of Medicine, Northwell Health at Mather Hospital, Port Jefferson, NY 11777, USA; 2Renaissance School of Medicine, Stony Brook University, Stony Brook, NY 11794, USA; 3Division of Cardiology, Albany Medical Center, Albany, NY 12208, USA; 4Division of Infectious Diseases, Northport Veterans Affairs, Northport, NY 11768, USA; 5Division of Cardiology, Department of Medicine, Stony Brook University Medical Center, 101 Nicolls Road, Health Science Center T-16, Rm. 080, Stony Brook, NY 11794, USA

**Keywords:** human immunodeficiency virus, cardiomyopathy, heart failure, HAART

## Abstract

The landscape of human immunodeficiency virus (HIV) epidemiology and treatment is ever-changing, with the widespread and evolving use of antiretroviral therapy (ART). With timely ART, people living with HIV (PLWH) are nearing the life expectancies and the functionality of the general population; nevertheless, the effects of HIV and ART on cardiovascular health remain under investigation. The pathophysiology of HIV-related cardiomyopathy and heart failure (HF) have historically been attributed to systemic inflammation and changes in cardiometabolic function and cardiovascular architecture. Importantly, newer evidence suggests that ART also plays a role in modulating the process of HIV-related cardiomyopathy and HF. In the short term, newer highly active ART (HAART) seems to have cardioprotective effects; however, emerging data on the long-term cardiovascular outcomes of certain HAART medications, i.e., protease inhibitors, raise concerns about the potential adverse effects of these drugs in the clinical course of HIV-related HF. As such, the traditional phenotypes of dilated cardiomyopathy and left ventricular systolic failure that are associated with HIV-related heart disease are incrementally being replaced with increasing rates of diastolic dysfunction and ischemic heart disease. Moreover, recent studies have found important links between HIV-related HF and other clinical and biochemical entities, including depression, which further complicate cardiac care for PLWH. Considering these trends in the era of ART, the traditional paradigms of HIV-related cardiomyopathy and HF are being called into question, as is the therapeutic role of interventions such as ventricular assist devices and heart transplantation. In all, the mechanisms of HIV-related myocardial damage and the optimal approaches to the prevention and the treatment of cardiomyopathy and HF in PLWH remain under investigation.

## 1. Introduction

Historically, as was popularized in Larry Kramer’s play “*The Normal Heart*”, human immunodeficiency virus (HIV) was initially thought to spare the heart; moreover, acquired immunodeficiency syndrome (AIDS) was considered to be a fatal entity of unknown etiology, particularly stigmatizing the gay community in the 1980s. During the following three decades, the developments in antiretroviral therapy (ART) have re-shaped HIV into a chronic condition, with near-normal life expectancy [1], and with causes of morbidity and mortality shifting from acute opportunistic infections to non-communicable diseases [2]. The evidence suggests that people living with HIV (PLWH) have a greater risk of developing cardiovascular disease (CVD) [2], especially heart failure (HF) [3] and stroke [4]. This effect is more prominent among those with a higher viral load and a lower CD4+ T cell count [5], and it is anticipated that by 2030, 73% of PLWH will be aged ≥50 and 78% will have CVD [6].

In 2019, the global prevalence of HIV was approximately 38 million, with 1.7 million newly acquired cases, compared to an incidence rate of 2.8 million in 1998 [7]. As of June 2020, 26 million people were accessing ART, which represents an increase from 25.4 million in 2019, and significant increase from 6.4 million in 2009 [7]. Despite the treatment effectiveness and the reduced incidence rate, HIV-associated CVD has tripled over the past twenty years globally, and is currently responsible for over 2.6 million annual disability-adjusted life-years, particularly in sub-Saharan Africa and Asian Pacific regions [2].

Early in the time course of HIV (i.e., in the 1980s and 1990s), heart failure among PLWH was primarily due to the direct viral effects and etiologies that are secondary to HIV infection (i.e., immunosuppression, opportunistic infections, secondary myocarditis, and nutritional deficiencies) [3,8,9,10]. Before the wider access to highly active ART (HAART), HIV-related cardiomyopathy was characterized principally by left ventricular (LV) systolic dysfunction and LV dilation, both of which had a poor prognosis. In the HAART era, overt cardiac involvement with systolic dysfunction in PLWH appears to have declined. Of note, this is mostly an empirical observation, as the evidence is limited [11]. Currently, it is thought that, because of the widespread use of HAART, CVD has become the most common cause death among PLWH [12]. With the decreasing rates of overt HF with a reduced ejection fraction (HFrEF) and a longer exposure to cardiometabolic risk factors and atherosclerosis, the etiology of myocardial involvement in PLWH has shifted towards coronary artery disease (CAD) and HF with preserved ejection fraction (HFpEF) [13].

In this article, we review the current evidence on the epidemiology of HIV-related HF and cardiomyopathy, discuss the pathophysiology, and outline the diagnostic and treatment options and the challenges that are associated with this entity.

## 2. Epidemiology of Heart Failure and Cardiomyopathy in HIV-Infected Patients

In a recent meta-analysis of over 125,000 adults with HIV (Figure 1), the pooled prevalence of LV systolic dysfunction was 12.3% (95% CI 6.4–19.7%), 12.0% (95% CI 7.6–17.2%) for dilated cardiomyopathy, 29.3% (95% CI 22.6–36.5%) for grades I to III diastolic dysfunction, and 11.7% (95% CI 8.5–15.3%) for grades II to III diastolic dysfunction [11]. The incidence rate and the prevalence of clinical HF were 0.9 per 100 person-years (95% CI 0.4 to 2.1) and 6.5% (95% CI 4.4% to 9.6%), respectively [11]. Furthermore, the pooled prevalence of pulmonary hypertension and right ventricular dysfunction was 11.5% (95% CI 5.5% to 19.2%) and 8.0% (95% CI 5.2% to 11.2%), respectively [11]. There was a trend for a lower prevalence of LV systolic dysfunction in studies reporting higher ART use and in the studies that were published more recently (Figure 2); nevertheless, for all of the estimates, a significant heterogeneity across the studies was observed [11]. Although generalizability is limited, another meta-analysis, which specifically addressed HF incidence, found that among nine million participants with >100,000 incident cases of HF, HIV infection was associated with a higher risk of HF (relative risk 1.80; 95% CI 1.51–2.15) [14].

## 3. Pathophysiology

Before the wider use of HAART, HIV-related heart failure was commonly a consequence of dilated cardiomyopathy, which was usually secondary to myocarditis and opportunistic infections. However, more recently, data seem to suggest that HIV-related cardiomyopathy is multifactorial, with causes including direct virally induced myocardial damage, with or without myocarditis, immune dysregulation, systemic and local inflammation, ischemic heart disease, pulmonary hypertension, HAART toxicity, and mental health disorders (Figure 3).

### 3.1. Direct HIV-Induced Myocardial Damage

Systolic dysfunction that is secondary to HIV has traditionally been correlated with the effects of the HIV-1 virus on the myocardium. In an experimental model of HIV infection, Hu-NSG mice of both male and female sexes developed LV diastolic dysfunction, with 25% exhibiting severe, restrictive diastolic dysfunction and mitral regurgitation four weeks after infection [15]. After eight weeks, there was evidence of LV systolic dysfunction, and after twelve weeks, 33% of the mice exhibited LV dyskinesia and dyssynchrony [16]. Histopathology revealed coronary microvascular leakage, fibrosis, and immune cell infiltration into the myocardium [16]. In line with these findings, in a cross-sectional study of contemporary patients on HAART, left atrial abnormalities and myocardial fibrosis were associated with an altered cardiac structure and consequent diastolic dysfunction [17].

### 3.2. Immune Dysregulation and Systemic Inflammation

In addition to direct myocardial damage, immune dysregulation and systemic inflammation have long been recognized as contributors to the development of cardiomyopathy in PLWH. Low CD4+ T cell count has repeatedly been associated with an increased risk of dilated cardiomyopathy and HF in PLWH [18,19]. In a landmark prospective study, inflammatory infiltrates and staining for major histocompatibility complex class I antigens was common among PLWH with dilated cardiomyopathy [18]. Additionally, patients with HIV-related dilated cardiomyopathy demonstrate a higher intensity of tumor necrosis factor-alpha (TNF-α) and inducible nitric oxide synthase staining on an endomyocardial biopsy compared to patients with idiopathic forms of dilated cardiomyopathy. The staining intensity is inversely correlated with the CD4+ T cell count [17].

Despite the beneficial effects of ART, persistent immune activation and systemic inflammation is evident in PLWH. Among approximately 400 patients with HIV from five African countries, the levels of several biomarkers of immune activation decreased after ART initiation. High pre-ART biomarker levels strongly predicted the residual immune activation after starting ART; the levels of C-X-C chemokine ligand 10, lipopolysaccharide-binding protein, C-reactive protein, soluble CD163, and soluble scavenger receptor CD14 were higher during ART than in the healthy controls, all of which suggest microbial translocation, persistent inflammation, and monocyte/macrophage activation [20].

The role of inflammation in heart disease and HF in PLWH is highlighted by the findings in “elite controllers”, i.e., patients with HIV infection with an undetectable HIV viral load despite no ART. In a study by the US-based HIV Research Network, after adjusting for demographic and relevant clinical variables, the elite controllers had over 3-fold higher rates of cardiovascular hospitalizations vs. the medically controlled patients [21]. AIDS-defining infections were rare among the elite controllers (2.7%); in contrast, cardiovascular hospitalizations occurred most frequently in this group (31.1%). Lastly, chest pain, coronary artery disease, and HF were the most common diagnoses leading to hospitalization [21].

Systemic inflammation in HIV infection does have myocardial effects. In a nonhuman primate model of HIV, for example, the TNF- α inhibitor etanercept prevented antigenic stimulation, which otherwise induced a dilated cardiomyopathy phenotype [22]. In patients with acute HIV infection, the levels of N-terminal B-type natriuretic peptide and troponin T are elevated [23]. The decreases in these biomarkers following viremic control track the partial normalization of the markers of systemic inflammation [23].

In all, the introduction of potent ART appears to control the overt viral effects and the intense systemic inflammation in PLWH, leading to a chronic state of low-level inflammation. In turn, this has caused a shift in the HIV-associated cardiomyopathy phenotype from overt left ventricular systolic dysfunction to cardiomyopathy with relatively preserved systolic function.

### 3.3. Myocardial Inflammation and Fibrosis

Despite the shifting prevalence from the predominance of HFrEF to HFpEF, detailed cardiac imaging studies still demonstrate persistent myocardial changes in PLWH, irrespective of the adherence to ART regimens. Cardiac magnetic resonance (CMR) data, for example, have shown evidence of both diffuse and focal myocardial fibrosis, elevated inflammatory parameters, and subclinical LV dysfunction with occasionally impaired systolic deformation, even in asymptomatic PLWH undergoing ART with adequate viral control (<200 copies/mL) [24]. Furthermore, in an observational imaging study, PLWH had significantly larger myocardial mass, with higher rates of fibrosis, lower LV ejection fraction, and a lower peak diastolic strain rate compared to the healthy controls [25]. In the characterizing heart function on antiretroviral aherapy (CHART) study, PLWH with diastolic dysfunction had a significantly higher prevalence of focal myocardial fibrosis compared to those without diastolic impairment (5.3% vs. 19.0%, respectively) [13].

### 3.4. Pulmonary Hypertension

The prevalence of pulmonary hypertension (PH) in the general global population is approximately 1% [26]. HIV-associated PH was first described in 1987 and has been increasingly recognized as a complication of HIV [27]. A recent meta-analysis of 42,642 PLWH had a PH prevalence of 8.3% in adults and 14.0% in adolescents, based on transthoracic echocardiographic data, despite significant heterogeneity between the studies [26]. Based on pathophysiology and clinical presentation, PH is classified into the following five groups: group 1, due to pulmonary arterial hypertension; group 2, due to left-sided heart disease; group 3, due to lung disease or hypoxia; and group 4, which is secondary to chronic thromboembolic PH and other pulmonary artery obstructions. HIV-associated PH is classified in group 1—pulmonary arterial hypertension (PAH) [26].

Histologically, HIV-associated PAH is indistinguishable from idiopathic PAH. The features of vascular remodeling (i.e., the hypertrophy of the tunica media, the hypertrophy of the muscular and elastic arteries, the dilation and intimal atheroma of the elastic pulmonary arteries, as well as characteristic plexiform lesions) are present in HIV–PAH and idiopathic variants alike [27]. Although the mechanisms of HIV-associated PAH are unclear, the direct viral effects on the vascular endothelium as a causal mechanism are not supported, as viral genetic material has not yet been found in the pulmonary vascular tissue [28]. The HIV viral proteins (e.g., the gp 120 envelope glycoprotein, the Nef protein, and the transcription-Tat protein) however, seem to play a key role in PAH-associated vascular remodeling by causing endothelial cell proliferation, inflammation, oxidative stress, and the deregulation of apoptosis [27,29]. Interestingly, a recent study evaluating the efficacy of statins on HIV–PAH in animal models determined that rate of the simian immunodeficiency virus (SIV)-associated PAH showed a decrease of 14.3% when healthy macaques were treated with atorvastatin prior to SIV infection, which was likely due to the amelioration of the inflammatory reponse during the post-acute phase [29]. Chronic inflammation, intravenous cocaine use (which promotes the proliferation of pulmonary artery smooth muscle cells synergistically with HIV-Tat), and herpesvirus 8 have also been suggested to contribute to the pathogenesis of PAH [28].

Clinically, in PLWH PAH increases the risk of mortality; namely, the causes of death that are attributed to PAH complications include fatal arrhythmias leading to sudden cardiac death and right heart failure (57–71%) [27]. However, right ventricular dysfunction in PLWH has been previously suggested to be a separate entity from pulmonary hypertension [30]. In a recent small study with 50 asymptomatic PLWH and 25 controls without echocardiographic findings or clinical signs of PH, PLWH had a significantly higher free wall thickness of the right ventricle, a higher right ventricular end-diastolic diameter, and a greater right atrial area and pulmonary arterial diameter compared to the control group. These findings suggest that right heart failure alternations can be present without PH [31].

### 3.5. Cardiometabolic Risk Factors and Ischemic Heart Disease

Cardiometabolic risk factors are prevalent among PLWH. In a recent meta-analysis pooling together cardiovascular risk factors and scores from 39 studies [32], the prevalence of moderate-to-high cardiovascular risk among PLWH was assessed with nine different scores, was 20.4% (95% CI 16.8–24.3). The most prevalent risk factors were dyslipidemia (39.5%), smoking (33.0%), hypertension (19.8%), and diabetes mellitus (7.2%) [32]. In view of this, and the long-standing HIV-associated inflammation, atherogenesis in PLWH is likely to be multifactorial. In a meta-analysis of coronary computed tomography data from 1229 PLWH and 1029 healthy controls, the rates of non-calcified coronary plaques were higher in the HIV group (58% versus 17%; OR 3.26, 95% CI 1.30–8.18), although the prevalence of calcified coronary plaques did not differ between the groups [33]. A similar study focusing on the pathophysiologic mechanisms of cardiac ischemia in PLWH found that myocardial oxygen supply and demand imbalance, rather than atherothrombotic disease, was found to be related to acute myocardial infarction events in up to half of the PLWH in another study [34].

The high prevalence of cardiometabolic risk factors and the persistent inflammation in PLWH also translates to higher rates of major cardiovascular events. In a meta-analysis of over 800,000 PLWH, the relative risk for cardiovascular disease was 2.16 (95% CI 1.68–2.77) generally, and 1.79 (95% CI 1.54–2.08) for myocardial infarction, specifically, compared to individuals without HIV [2]. Moreover, viral suppression may not eliminate the excess risk. In the Veterans Aging Cohort Study, the veterans with HIV had a 48% higher risk of myocardial infarction compared to the healthy controls (95% CI 0.27–0.72) after adjustment for risk factors including illicit substance use and other comorbidities [35]. Importantly, the risk remained higher among PLWH with a HIV-1 RNA level < 500 copies/mL in subgroup analyses (hazard ratio 1.39; 95% CI 1.17–1.66) [35]. Lastly, hypotheses exist suggesting that PLWH may be at higher risk of developing HF after myocardial infarction compared to people without HIV. In a French hospital-based registry, HIV infection was an independent predictor of incident HF and hospitalization for HF one year after myocardial infarction [36].

### 3.6. HIV Medications

The evidence that is currently available suggests that the use of HAART may be of overall benefit to the cardiomyopathic process in HIV. For example, the newer antiretroviral medications, such as the integrase inhibitors dolutegravir and raltegravir, and the C-C chemokine receptor 5 antagonist maraviroc have been shown to improve the surrogate markers of atherosclerosis (i.e., the flow-mediated vasodilation and intimal-medial thickness in the carotid artery) and to favorably alter lipid levels, which may, therefore, decrease the risk of CVD in PLWH [37].

Nevertheless, certain HAART regimens may have adverse cardiovascular effects in the long term. Abacavir, which is a widely prescribed nucleoside reverse transcriptase inhibitor, was associated with a higher risk of myocardial ischemia and infarction in some observational studies, which was attributed largely to an increase in vascular inflammation and platelet reactivity; nevertheless, a meta-analysis showed no significant ischemic risk [38,39]. Current HAART guidelines in the United States recommend judicious use or the avoidance of abacavir entirely in PLWH who have, or who are at a high risk for developing, CVD [34].

Conflicting data also exist regarding other HAART drugs, such as protease inhibitors. A recent retrospective study of 394 PLWH with HF found that the use of protease inhibitors was associated with dyslipidemia, diabetes, coronary artery disease, a decreased LV ejection fraction, and an overall increase in the 30-day readmission rate and cardiovascular mortality [40]. Biochemical data reveal that numerous HIV protease inhibitors downregulate the enzyme ZMPSTE24, which is responsible for the processing of the pro-cardiomyopathic protein prelamin A [41]. Since the accumulation of prelamin A has been documented in cardiac biopsies of PLWH, the inhibition of the ZMPSTE24 has been a proposed mechanism by which the protease inhibitors may increase the risk of acquired cardiomyopathy in this patient group. Importantly, emerging data suggest that newer protease inhibitors, such as atazanavir and darunavir, are likely to be safer to use in PLWH with CVD compared to earlier-generation medications [42].

Lastly, certain antiretroviral agents in the nucleoside reverse transcriptase inhibitor are known to have some degree of mitochondrial toxicity [43]. Zidovudine, clevudine, and iodenosine, for example, have been reported to produce defects in mitochondrial DNA replication and energetics [44,45,46,47], which may alter myocardial function. In addition, HIV infection per se can cause mitochondrial toxicity, as has been demonstrated in experimental animal models [48]. Nevertheless, robust evidence is lacking, and further work is required in order to investigate the effects of potential HIV- and antiretroviral therapy-induced mitochondrial changes in cardiovascular function.

### 3.7. Depression

An important, yet understudied, facet of heart failure care for PLWH is the consideration of mental health. The psychosocial influence on cardiovascular disease in general has been recognized for some time; however, quantitative studies on the impact of mood disorders on HF in HIV had not been published until recently. A 2015 cohort study of 81,427 veterans found that major depressive disorder was an independent predictor of HF in PLWH after a mean follow-up period of 5.8 years (HR 1.68; 95% CI 1.45–1.95). Importantly, the use of antidepressant medications was shown to significantly mitigate the risk of heart failure (HR 0.76; 95% CI 0.58–0.99) [49]. Although data on the causal mechanisms are limited, this study highlights the importance of psychosocial variables on heart function and indicates a need for additional work on this topic.

### 3.8. HIV-Related HF and Cardiomyopathy in Pregnancy

Although peripartum cardiomyopathy is a unique clinical entity that has been described at length in the literature, very little is known about the relationship between HIV infection and consequent HF or cardiomyopathy in the setting of pregnancy. Only limited case series and few observational studies to date have included pregnant PLWH; thus, the question of whether or not cardiomyopathy and/or HF in pregnant PLWH resembles classic peripartum cardiomyopathy or if it is a distinct pathology remains unanswered. Mandal and colleagues have described a single case of comorbid HIV and cardiomyopathy in a 31-year-old pregnant woman, in which imaging demonstrated a typical phenotype of peripartum dilated cardiomyopathy, which was poorly responsive to diuresis, ACE inhibitors, and inotropes, and eventually resulted in a fatal ventricular tachycardia [50]. In a prospective study of 80 African women with peripartum cardiomyopathy, 34% had comorbid HIV; no difference in mortality or LV ejection fraction was noted between patients with HIV compared to those without HIV [51]. Lastly, a comparative imaging study showed no difference in the echocardiographic measurement of the cardiac output, the mitral valve E/E’ ratio, the LV mass index, the LV end-diastolic pressure, the LV ejection fraction, or the LA volume index between a small sample of Ugandan woman with HIV (N = 41) and those without HIV (N = 41) [52].

Similarly, literature on the effects of comorbid peripartum cardiomyopathy and HIV on fetal development is scarce. The same study of Ugandan women compared fetal indices, including the gestational age at delivery, the birth weight, and the AGPGAR scores, and found no differences between the children who were born to PLWH and women without HIV [52]. With respect to the fetal cardiac changes in response to HARRT, the data are conflicting. De la Calle and colleagues, for example, found no significant echocardiographic changes in HAART-exposed fetuses [53]; however, Garcia-Otero et al., found increased myocardial mass in the fetuses whose mothers’ were taking zidovudine during pregnancy [54]. Overall, this specific area is heavily under-researched, and no definitive conclusions can be drawn.

## 4. Diagnosis of Cardiomyopathy and Heart Failure in Patients with HIV

### 4.1. Clinical Presentation

The clinical manifestations of HF and cardiomyopathy in PLWH are heterogenous. The acute presentations of HIV-related cardiovascular emergencies include acute coronary syndromes, pericardial effusions or tamponade, opportunistic infective endocarditis, pulmonary embolism, and cardiac rhythm abnormalities [55]. Pericardial effusions have been found to be up to three times more likely [25], venous thromboembolism two to ten times more likely (especially if the CD4+ T cell count is low) [56], and sudden cardiac death twice as likely [55,57] in PLWH compared to their healthy counterparts. When managing patients acutely, chest pain, dyspnea, hypoxia, and/or arrythmias should prompt comprehensive cardiac and infectious workups, as new-onset or worsening LV dysfunction that are secondary to HIV-related disease (i.e., lymphopenia, Kaposi sarcoma, lymphoma, etc.) may present anywhere on a severity spectrum, from mild symptoms to overt hemodynamic collapse.

### 4.2. Biomarkers

Although the exact pathophysiologic mechanisms are not yet fully understood, the existing evidence suggests that several biomarkers of systemic and myocardial changes that are present in the general population are also associated with HIV-induced cardiac dysfunction. Despite the current limited clinical application, a moderately powered study found that the categorization of PLWH by serum biomarkers may have a role in defining the treatment goals and assessing the prognosis [58]. A summary of the important biomarkers that are related to HIV cardiomyopathy and HF is presented below.

Galectin-3 has been shown to be overexpressed in PLWH and is associated with an increased risk of HF and mortality [59,60]. Another fibrotic marker is growth differentiation factor 15, which is an independent predictor of all-cause mortality in PLWH [61,62]. Similarly, soluble ST2, which plays a role in diastolic dysfunction and is a predictor of mortality in HFpEF patients, has also been found to be associated with the structural changes in the myocardium of PLWH [63,64]. Regarding the systemic markers, PLWH with diastolic dysfunction had higher levels of carboxy-terminal telopeptide of collage type I, interleukin-6, oxidized low-density lipoprotein, and D-dimer compared to PLWH without diastolic failure [13,58]. Furthermore, limited evidence suggests that elevated N-terminal pro-B-type natriuretic peptide (NT-proBNP) is associated with higher rates of pulmonary hypertension and cardiovascular mortality [58,65]. Of note, over 60 monocyte-derived inflammatory markers were found to be associated with diastolic dysfunction, and were also found to be higher in women with HIV undergoing ART compared to the control group [37]. The potential mechanistic roles in HIV-induced LV dysfunction of matrix metalloproteinases [66,67] and lipids, such as acylcarnitine, triacylglycerols, and diacylglycerols, have been investigated [68], but more robust data are needed in order to fully understand these.

## 5. Treatment

### 5.1. Medical Therapy

To date, there have been no published randomized trials of HF medications in PLWH, and as such, optimal pharmacotherapy for HIV-related HF remains uncertain. HIV serostatus data are not usually collected in clinical trials; therefore, the data that are derived from retrospective analyses and case series or from extrapolation from the general population are the main drivers of therapeutic recommendations. Nevertheless, a crucial aspect of therapy includes the treatment of cardiovascular comorbidities, including hypertension, diabetes mellitus (DM), and dyslipidemia.

Although the data are sparse, it is known that the HIV-1 virus stimulates renin production by the CD4+ T cells, which contributes to a renin–angiotensin–aldosterone system (RAAS)-mediated hypertension; consequently, the first-line antihypertensives in PLWH are angiotensin converting enzyme (ACE) inhibitors and angiotensin receptor blockers (ARBs) [37]. Although generalizability is limited, an observational study of heart failure patients in Botswana and Uganda (the prevalence of HIV was 33.9% and 18.6%, respectively) showed similar responses to ACE or ARB therapy between PLWH and patients without HIV [69,70]. Interestingly, a retrospective study that was derived from the Antihypertensives in Obesity Management study cohort found that beta blockers had increased risk of incident cardiovascular disease compared to ACE/ARBs (HR [95% CI]: 1.90 [1.24–2.89]) in PLWH who had hypertension [71]; importantly HF and cardiomyopathy were not studied specifically in this paper. Currently, a phase II randomized controlled trial, ENCHANTMENT HIV, is recruiting participants in order to investigate the effects of the angiotensin receptor neprilysin inhibitor sacubitril/valsartan on cardiac imaging endpoints, with the long-term goal to prevent the transition from HIV-related Stage B HF to clinical Stage C HF (NCT: 04153136).

While specific guidelines for the treatment of DM in PLWH do not formally exist, traditional medical therapies have cardioprotective effects that may benefit PLWH as well. Previous studies have proposed several mechanisms (e.g., the inhibition of pro-inflammatory cytokines and the modification of T cell-specific protein kinases) by which metformin, which is a frequently used antihyperglycemic drug, may theoretically increase the CD4+ T cell counts [37]. The recent LILAC clinical trial has strengthened this evidence base, as the use of metformin showed a reduction in colonic CD4+ T cell infiltration and a reduction in surrogate markers of HIV viral transcription [72]. In addition to metformin, newer DM medications, such as sodium–glucose cotransporter 2 inhibitors (SGLT2i), have also been cited as theoretically efficacious in PLWH with HF, secondary to their known benefits of reducing catecholamine-mediated changes in the myocardial architecture and consequent myocardial fibrosis [37]. Several ongoing randomized controlled trials are addressing the efficacy of SGLT2i in PLWH, as outlined in Section 5.5 below.

Lastly, antilipemic agents, such as statins, are known to have some immunomodulatory effects; however, data are lacking with respect to the effects of statins on cardiovascular function in PLWH. The ongoing REPREIVE trial aims to investigate the role of pitavastatin on the incidence rate of cardiovascular mortality [73].

### 5.2. Implantable Devices

The use and the effects of implantable cardiac devices in PLWH with cardiomyopathy have not been investigated thoroughly. Previous authors have concluded that PLWH are less likely to receive an implantable defibrillator or cardiac resynchronization therapy device, which is likely because of the misbelief that PLWH have shorter life expectancies, or due to concerns regarding potential infectious complications [74]. Although there may be some validity to these concerns (i.e., a recent study reported higher rates of bacteremia despite HAART in PLWH compared to the general population), other explanations, including stigma and similar psychosocial variables, may play a role in the decision-making process regarding implantable devices [75].

### 5.3. Transplant and Circulatory Support Devices

In the past, high mortality and immunosuppressive concerns have made HIV infection generally considered a contraindication for heart allotransplanttion [76]. Importantly however, data on the prognosis of PLWH who have undergone heart transplantation in the HAART era are limited [77]. Whereas earlier studies of cardiac transplantation in PLWH demonstrated poor outcomes [78], more recent literature has not shown an increase in organ rejection or worsening of immunosuppressive sequalae among these patients [79,80]. Overall, since the consensus data suggest that PLWH may have cardiovascular improvement if undergoing HAART, selected patients may be reasonable candidates for a heart transplant [77,81].

The data on the benefit of mechanical circulatory support devices in PLWH are limited. However, there have been acceptable outcomes in case series (72% survival at 24 months post-procedure) in PLWH who were left ventricular assist device (LVAD) recipients, with no significant adverse events that were attributed to the HIV infection itself [82,83]. Previous work has identified that many centers view a HIV positive status as a contraindication to LVAD use, leading to a disproportionately lower number of PLWH (who are otherwise good candidates) receiving LVADs [83]. Although the need for more data is clear, the recent data support the consideration of PLWH for advanced therapy, including transplant and mechanical circulation, to treat HIV-associated cardiomyopathy.

### 5.4. HAART and Immune Therapy

HAART is known to have a multi-factorial effect on the myocardium. While some case reports have found regression of cardiomyopathy in adults [84] and children [85], a recent retrospective study found that protease inhibitors were associated with worse HF outcomes among PLWH [40]. Furthermore, the START trial showed a reduction in mortality and major HIV/AIDS-related morbidity in patients who started HAART at initial CD4+ T cell counts of >500 compared to those with lower initial CD4+ counts [34]. Importantly, there was a numeric, but non-statistically significant, reduction in major cardiovascular events in patients who were enrolled in the START trial, which limits the generalizability of results, but nevertheless reveals important information about the potential benefits of HAART-based immunomodulation in CVD [86,87]. Furthermore, a retrospective study of 49 children with a HIV infection found a notable decrease in the LV wall thickness and peak wall stress, and an improved LV contractility and fractional shortening after treatment with intravenous immunoglobulins [88]. The administration of intravenous immunoglobulin may, therefore, provide some therapeutic benefit that is secondary to the inhibition of inflammatory cytokines, such as TNF and interleukins, but robust data remain sparse [3]. Nevertheless, the role of immune therapy in HIV-related HF remains understudied. An important question that remains unanswered and merits further exploration is the mechanistic role of HAART in the reversal of HIV-related cardiomyopathy.

### 5.5. Therapeutic Perspectives

A number of ongoing clinical trials are addressing the potential benefit of newer drugs for cardiomyopathy and HF in PWLH. These include the mechanistic ENCHANTMENT HIV trial with sacubitril/valsartan to halt the progression of HIV-associated cardiac structural changes, as well as another Phase II study to assess the effects of the SGLT2i canagliflozin on the progression of prediabetes to diabetes, including several cardiometabolic secondary endpoints (NCT: 05135039), and a 48-week randomized, open-label trial of a three-drug (dolutegravir/abacavir/lamivudine) vs. a two-drug (dolutegravir/lamivudine) antiretroviral regimen. This will include a comprehensive assessment of changes in cardiac risk, the composition and calcification of cardiac tissue, and changes in the body composition, the metabolism, inflammation, and coagulation (NCT: 04904406).

## 6. Prognosis of Heart Failure in HIV-Infected Patients

The data on the prognosis of HIV-associated HF are limited. Although HF prognosis in the general population has improved in the past decades, the outcomes of HIV-associated HF are still unclear. In a retrospective cohort study of Veterans Health Affairs data from 2000 to 2018, compared to veterans with HF but without HIV, their HIV-positive counterparts had a higher one-year mortality (30.7% vs. 20.3%), HF admission (21.2% vs. 18.0%), and all-cause admission (50.2% vs. 38.5%) rates, even after adjustment for demographic variables [89]. The outcomes were worse for PLWH with a lower CD4+ T cell count, a higher viral load, and a lower LV ejection fraction, signifying the importance of HIV infection control in the HF outcomes for these patients [89].

However, as discussed previously, the specific type of ART may affect the prognosis of HIV-associated HF. In a single-center study of 394 PLWH who were hospitalized with HF, higher rates of coronary artery disease, diabetes mellitus, hyperlipidemia, higher pulmonary artery systolic pressures and lower LV ejection fraction were associated with use of protease inhibitors [40]. Furthermore, the use of protease inhibitors was associated with increased 2-year cardiovascular-cause mortality (35% vs. 17%; *p* < 0.001) and a higher 30-day HF readmission rate (68% vs. 34%; *p* < 0.001) [40].

Currently, cardiac diseases account for more than 25% of deaths among PLWH, compared to less than 10% before the use of HAART. Furthermore, echocardiographic evidence of cardiomyopathy and symptomatic HF are associated with 4.0 and 6.5 times greater risk for death, respectively [90,91]. In a subset of patients with available echocardiograms, a high prevalence of structural and functional abnormalities was observed [91]; nevertheless, data are too sparse to clearly define the role of cardiac imaging in the screening and the prognosis of HIV-related HF and cardiomyopathy.

Importantly, the existing data have demonstrated that the risk of sudden death may be up to four time more likely in PLWH with a significant viral load and/or CD4+ T cell count of <200 compared to PLWH without lymphopenia and an undetectable viral load [92]. Although many potential explanatory mechanisms exist, some authors have found that noncardiac causes of sudden death, including fatal overdose and thromboembolic events, are higher than were previously estimated among PLWH [93]. These findings are critical in the changing landscape of comprehensive HIV treatment and should be explored further.

Lastly, the functional outcomes of HIV-related HF and cardiomyopathy have been recognized. The CHART study has affirmed the previous results indicating that diastolic dysfunction in PLWH was associated with a greater frequency of symptoms and higher rates of physical limitation compared to the controls [13]. Related to the functional outcomes, the readmission rates for HF and sequelae have also been shown to be higher in PLWH. In a retrospective observational study of veterans with HIV, the hospitalizations were more frequent compared to the healthy controls, and as expected, the mortality was higher among a subset of this cohort with a lower ejection fraction and CD4+ T cell count and a higher burden of viremia [89]. Similarly, a propensity score-matched analysis of heart failure patients found higher rates of rehospitalization for HF exacerbations among PLWH compared to healthy controls [94].

## 7. Disparities in Heart Failure Risk and Care among HIV-Infected Persons

In a large administrative database comprising 36,400 patients with HIV and over 12 million control individuals [95], although the prevalence of HF was higher in the older PLWH, the relative risk that is associated with HIV was highest in the younger people and in women. In addition, PLWH with HF were less likely to receive antiplatelet drugs, statins, diuretics, and renin–angiotensin system antagonists compared to their healthy counterparts. Care by a cardiologist was associated with more frequent use of antiplatelets, statins, renin–angiotensin system antagonists, and diuretics among HIV-infected patients with HF [95].

Women with HIV have an increased risk for HF [96], and in the United States, they have increased rates of hospitalization for HF [97]. Vulnerability to HF in women with HIV might in part be related to an increased immune activation due to HIV [98]. Women with HIV are also more likely to have myocardial fibrosis, decreased diastolic function, and elevated markers of inflammation [99]. Importantly, women with HIV are less likely to receive interventions that prevent and treat cardiovascular disease compared to men with HIV at comparable cardiovascular risk, highlighting a critical need to re-evaluate the gendered approach to cardiovascular care [100].

## 8. Conclusions

Our understanding of HIV-associated cardiomyopathy has evolved, but it is still incomplete. The widespread use of HAART has changed the phenotype of HIV-related HF from a severe, dilated form of cardiomyopathy to one of less severe LV systolic dysfunction and varying degrees of diastolic dysfunction. Furthermore, the significance of diastolic impairment in PLWH has not been fully defined yet, and further research is needed in order to determine the best practices for diagnosis and treatment. Although a growing evidence base suggests the cardioprotective role of HAART, such benefits may be curtailed in the long-term due to adverse effects on the myocardium itself or on cardiovascular risk factors, such as dyslipidemia. These effects may vary from class to class of antiretroviral drugs, and further work is needed in order to define this more completely. In conclusion, the best therapeutic approach for patients with HIV-associated cardiomyopathy remains unclear. In light of the prevalence of cardiac abnormalities among PLWH and the scarcity of evidence-based guidelines on how to best diagnose and treat these patients, further research on this topic is imperative.

## Figures and Tables

**Figure 1 jpm-12-01760-f001:**
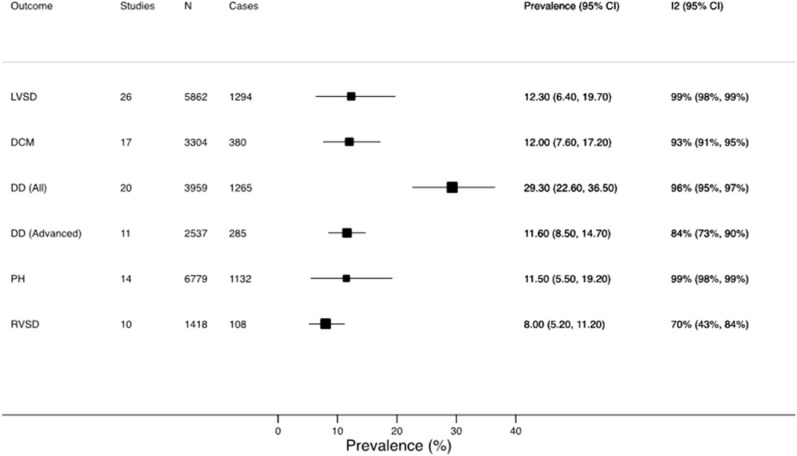
Pooled prevalence of cardiac impairment in adults with HIV infection. Reproduced with permission from: Erqou et al., J Am Coll Cardiol HF 2019;7:98–108 [11]. CI: confidence interval; DCM: dilated cardiomyopathy; DD: diastolic dysfunction; LVSD: left ventricular systolic dysfunction; PH: pulmonary hypertension; RVSD: right ventricular systolic dysfunction.

**Figure 2 jpm-12-01760-f002:**
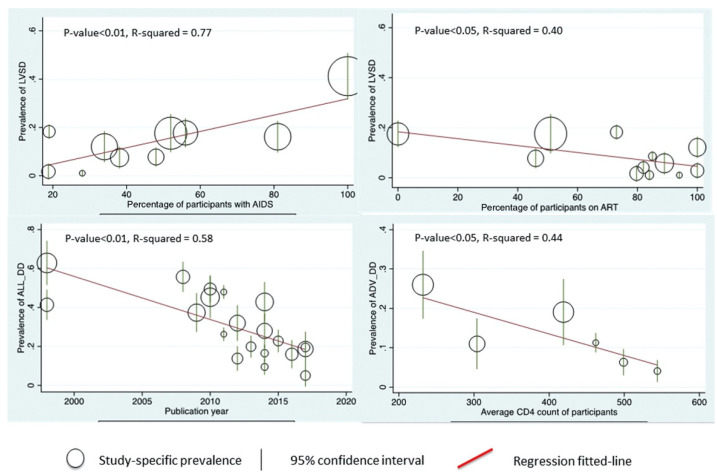
Meta-regression analysis of left ventricular systolic dysfunction (LVSD) and diastolic dysfunction (DD) by various characteristics. Reproduced with permission from: Erqou et al., J Am Coll Cardiol HF 2019;7:98–108 [11]. ALL_DD: All diastolic dysfunction; ART: antiretroviral therapy; ADV_DD: advanced diastolic dysfunction; CD4: CD4 T cell count; CI: confidence interval; DD: diastolic dysfunction; LVSD: left ventricular systolic dysfunction.

**Figure 3 jpm-12-01760-f003:**
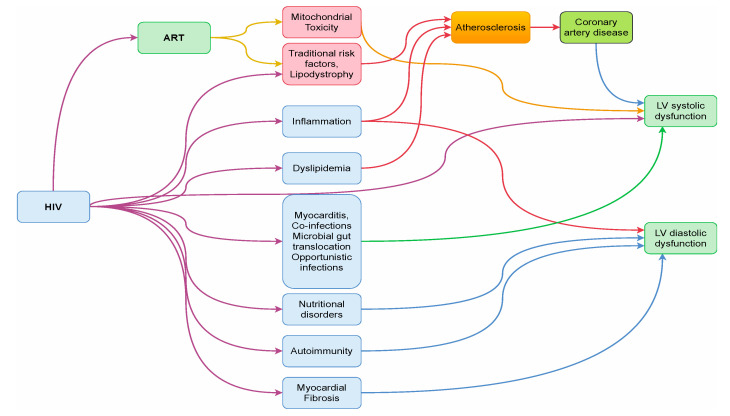
Proposed mechanisms of systolic and diastolic cardiac dysfunction in HIV-infected persons. ART = Antiretroviral therapy; LV = left ventricular.

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
