# Peer review of "Human Immunodeficiency Virus Infection-Associated Cardiomyopathy and Heart Failure"

_jpm, 2022, doi:10.3390/jpm12111760_

Round 1

Reviewer 1 Report

This review by A. Papamanoli and colleagues summarized HIV infection associated cardiomyopathy and HF. The manuscript is a good read and writing, presentation are praiseworthy. Here are my concerns.

1. Fonts of Figure 1 needs to be increased. 

2. In Figure 3 scheme ART related mitochondrial toxicity is relevant in heart as described in line 275-280. However, HIV itself cause Mitochondrial toxicity which might lead to cardiac dysfunctions reported at least in animal models by a Temple University, Philadelphia group on 2019. 

3. HIV related cardiac dysfunction in pregnant women and fetus born with HIV related cardiac remodeling could be included. 

Author Response

This review by A. Papamanoli and colleagues summarized HIV infection associated cardiomyopathy and HF. The manuscript is a good read and writing, presentation are praiseworthy. Here are my concerns.

We thank the reviewer for the positive feedback. Please see below responses to specific concerns.

  1. Fonts of Figure 1 needs to be increased. 

We agree, but unfortunately, we do not have access to the original figure in vector format, and this is the highest quality available in the journal website. We tried to enhance the figure using various tools, but the outcome is not satisfactory, so we decided to leave as is. However, we upsized the figure within the journal style limits.

  1. In Figure 3 scheme ART related mitochondrial toxicity is relevant in heart as described in line 275-280. However, HIV itself cause Mitochondrial toxicity which might lead to cardiac dysfunctions reported at least in animal models by a Temple University, Philadelphia group on 2019. 

We agree with the reviewer. We therefore updated Figure 3 to include this important link and we have updated the references accordingly. We have appended here the updated Figure 3 for convenience. We could not locate the Temple reference; however, we cited a comprehensive review on animal models of HIV-induced mitochondrial toxicity from the Emory group, which has substantial experience with HIV pathophysiology (Koczor et al, Methods 2010;51:399-404).

  1. HIV related cardiac dysfunction in pregnant women and fetus born with HIV related cardiac remodeling could be included. 

We thank the reviewer for this suggestion. We have included a new subsection in the Pathophysiology section (section 3.7) which now reads:

“3.7 HIV-related HF and cardiomyopathy in Pregnancy

Although peripartum cardiomyopathy is a unique clinical entity that has been described at length in the literature, very little is known about the relationship between HIV infection and consequent HF or cardiomyopathy in the setting of pregnancy. Only limited case series and few observational studies to date have included pregnant PLWH, thus the question of whether or not cardiomyopathy and/or HF in pregnant PLWH resembles classic peripartum cardiomyopathy or is a distinct pathology remains unanswered. Mandal and colleagues describe a single case of comorbid HIV and cardiomyopathy in a 31-year-old pregnant woman: imaging demonstrated a typical phenotype of peripartum dilated cardiomyopathy, which was poorly responsive to diuresis, ACE inhibitors, and inotropes, and eventually resulted in a fatal ventricular tachycardia [50]. In a prospective study of 80 African women with peripartum cardiomyopathy, 34% had comorbid HIV; no difference in mortality or LV ejection fraction was noted between patients with HIV versus without [51]. Lastly, a comparative imaging study showed no difference in echocardiographic measurement of cardiac output, mitral valve E/E’ ratio, LV mass index, LV end-diastolic pressure, LV ejection fraction, or LA volume index between a small sample of Ugandan woman with (N=41) versus without HIV (N=41) [52].

Similarly, the literature on effects of comorbid peripartum cardiomyopathy and HIV on fetal development is scarce. The same study of Ugandan women compared fetal indices including gestational age at delivery, birth weight, and AGPGAR scores, and found no differences between children born to women PLWH versus women without HIV[52]. With respect to fetal cardiac changes in response to HARRT, data are conflicting. De la Calle and colleagues for example, found no significant echocardiographic changes in HAART-exposed fetuses [53], however Garcia-Otero et al., found increased myocardial mass in fetuses whose mothers’ were taking zidovudine during pregnancy [54]. Overall, this specific area is heavily under-researched, and no definitive conclusions can be drawn.”

Reviewer 2 Report

In this article authors discussed current evidences on the epidemiology, pathophysiology, diagnosis and treatment of HIV-related  cardiomyopathy.

In my opinion the paper is well-written ad rather complete in each paragraph.

I have oonly few suggestions:

Medical Therapy: this paragraph should  be implemented: the current role of betablockers in HIV-related cardiomiopathy should be described . Moreover available data on the effcts of sacubitril-valsartan use in HIV-related cardiomiopathy should be reported

I suggest to add a new short paragraph about the therapeutic perspectives with short mentions on ongoing clinical trials involving newest drugs (for example SGLT2 inhibitors)

Author Response

In this article authors discussed current evidences on the epidemiology, pathophysiology, diagnosis and treatment of HIV-related cardiomyopathy.

In my opinion the paper is well-written ad rather complete in each paragraph.

We thank the reviewer for the positive feedback.

I have only few suggestions:

  1. Medical Therapy: this paragraph should be implemented: the current role of betablockers in HIV-related cardiomyopathy should be described . Moreover available data on the effects of sacubitril-valsartan use in HIV-related cardiomyopathy should be reported

We thank the reviewer for this suggestion. We agree and have included updated information on the use of beta blockers and ARNIs in the treatment of HIV-related heart failure and cardiomyopathy. Treatment section (5.1) now includes the following information:

“Interestingly, a retrospective study derived from the Antihypertensives in Obesity Management study cohort found that beta blockers had increased risk of incident cardiovascular disease compared to ACE/ARBs (HR [95%CI]: 1.90 [1.24-2.89]) in PLWH who had hypertension [71]; importantly HF or cardiomyopathy was not studied specifically in this paper. Currently, a phase II randomized controlled trial, ENCHANTMENT HIV, is recruiting participants to investigate the effects of the angiotensin receptor neprilysin inhibitor sacubitril/valsartan on cardiac imaging endpoints, with the long-term goal to prevent transition from HIV-related Stage B HF to clinical, Stage C HF (NCT: 04153136).”

  1. I suggest to add a new short paragraph about the therapeutic perspectives with short mentions on ongoing clinical trials involving newest drugs (for example SGLT2 inhibitors)

We agree with the reviewer and therefore we have added a new subsection (5.5 Therapeutic Perspectives), where we summarize ongoing clinical trials with new agents in these patients:

“A number of ongoing clinical trials are addressing the potential benefit of newer drugs for cardiomyopathy and HF in PWLH. These include the mechanistic ENCHANTMENT HIV trial with sacubitril/valsartan to halt the progression of HIV-associated cardiac structural changes (NCT: 04153136) as discussed above, a phase II study to assess the effects of the SGLT2i canagliflozin on the progression of prediabetes to diabetes, including several cardiometabolic secondary endpoints (NCT: 05135039), and a 48-week randomized, open-label trial of a three-drug (dolutegravir/abacavir/lamivudine) vs. a two-drug (dolutegravir/lamivudine) antiretroviral regimen, which will include a comprehensive assessment of changes in cardiac risk, composition and calcification of cardiac tissue, and changes in body composition, metabolism, inflammation, and coagulation (NCT: 04904406).”

Reviewer 3 Report

I have reviewed the manuscript entitled 'Human Immunodeficiency Virus Infection-Associated Cardio- 2 myopathy and Heart Failure'.
The review is interesting and has a great value to be published.
As a diagnostic tool of HIV related diastolic failure,  new ECG scoring systems can be used please consider citing while adding this section 'A simple formula to predict echocardiographic diastolic dysfunction-electrocardiographic diastolic index'
Moreover, several scoring systems can be used when deciding who to implant devices. Please consider citing while adding a section to this part 'Comparison of mortality prediction scores in elderly patients with ICD for heart failure with reduced ejection fraction' and The value of C-reactive protein-to-albumin ratio in predicting long-term mortality among HFrEF patients with implantable cardiac defibrillators

Author Response

The review is interesting and has a great value to be published.

Thank you for the positive feedback.

  1. As a diagnostic tool of HIV related diastolic failure,  new ECG scoring systems can be used please consider citing while adding this section 'A simple formula to predict echocardiographic diastolic dysfunction-electrocardiographic diastolic index'

We thank the reviewer for the suggestion. However, this work is not specific to patients with HIV, and we have therefore opted not to include in the paper.

  1. Moreover, several scoring systems can be used when deciding who to implant devices. Please consider citing while adding a section to this part 'Comparison of mortality prediction scores in elderly patients with ICD for heart failure with reduced ejection fraction' and The value of C-reactive protein-to-albumin ratio in predicting long-term mortality among HFrEF patients with implantable cardiac defibrillator

Again, we feel that this work is not directly relevant to the topic of our work, namely HIV related heart failure and cardiomyopathy, and therefore decided not to include in the paper.

Round 2

Reviewer 3 Report

The authors did not administer the required changes.